# InterIDEAS: An LLM and Expert-Enhanced Dataset for Philosophical Intertextuality

## Abstract

The formation and circulation of ideas in philosophy have profound implications for pedagogical and scholarly practices. However, traditional analyses often depend on manual reading and subjective interpretation, constrained by human cognitive limits. To address these challenges, we introduce InterIDEAS, a pioneering dataset designed to bridge philosophy and natural language processing (NLP). By merging theories of intertextuality from literary studies with bibliometric techniques and recent LLMs, InterIDEAS enables both quantitative and qualitative analysis of the intellectual, social, and historical relations embedded within these difficult-to-interpret philosophical texts. This dataset not only enhances the study of philosophy but also contributes to the development of language models by providing a training corpus that challenges and enhances their interpretative capacity. InterIDEAS covers over 45,000 pages from key philosophical texts, spanning major thoughts and schools from 1750 to 1950, and features more than 3,150 writers. It manifests the mutual contribution between philosophy and NLP, laying the groundwork for future interdisciplinary research.

## 1 Introduction

Although philosophy seems to be produced independently by a few genius thinkers, ideas do not exist in a vacuum. Philosophers read, cite, and discuss each other. Thus, intertextuality—the relationship among different texts established by their referencing to or commenting on each other—is one of the most crucial ways to situate an idea in its epistemological, disciplinary, and social contexts. An adequate interpretation of even a single philosophical concept requires the reading of a vast collection of texts to understand with whom the philosopher(s) conversed, what sociohistorical incidents they responded to, and what intellectual foundation was evoked to establish their perspective.

Previous researchers have addressed intertextuality via bibliometrics (Hammarfelt, 2016; Glänzel & Schoepflin, 1999): by quantitatively analyzes citation entries, scholars can measure the relationships among texts and gain broad insights about a topic or even an entire discipline. However, directly extracting bibliographies from philosophy texts is not feasible in philosophy, unless we limit ourselves to a very specific domain and to texts produced in a narrow span of time (Ahlgren et al., 2015). First, the lack of standardized citation practices before the mid-twentieth century results in a wide variety of formats that automated systems struggle to interpret. Second, the density of philosophical writing imposes tremendous interpretative challenges for digitalization.

For instance, a typical intertextual case in philosophy may read as follows: *"The striving toward phenomenology was present already in the wonderfully profound Cartesian fundamental considerations; then, again, in the psychologism of the Lockean school; Hume almost set foot upon its domain, but with blinded eyes. And then the first to correctly see it was Kant, whose greatest intuitions become wholly understandable to us only when we had obtained by hard work a fully clear awareness of the peculiarity of the province belonging to phenomenology."* (Husserl & Moran, 2012, p.142) Many factors contribute to the obscurity of this passage: a series of names, references, and concepts are crammed into a narrow space; the author writes rhetorically; the author does not specify his opinion to each mentioned philosopher and expects readers to uncover logical connections throughout the passage based on their previous philosophical knowledge; moreover, seemingly unimportant words like "almost" and "only" radically alter the author's attitude. All this subtlety needs to be identified, organized, and analyzed through a specifically designed data extraction process.

While recent development in Large Language Models (LLMs) offers potential break-through—considering their effectiveness in summarizing, extracting, and discovering textual knowl-edge—several challenges remain. First, the density of philosophical writing as illustrated above still lies beyond usual NLP tasks and training corpora. Second, unlike fields such as medicine, law, or general literature, there are relatively few curated datasets in philosophy to serve as the groundwork for more nuanced information extraction and analysis. Third, there are only very few philosophical scholars with expertise in AI, and vice versa. The disciplinary gap hinders the application of LLMs to philosophical research.

Once realized, philosophical intertextuality will further offer significant insight to NLP. Its incorpo-ration to foundation models facilitates the latter to venture beyond surface-level text interpretation. By tackling with latent emotion, contextual dependencies, and abstract reasoning, LLMs improve their capacity in citation extraction, argument mining, and sentiment analysis. LLMs' successful application to philosophical intertextuality will further imply their potential in assisting research in other humanistic disciplines, like literature and law, where the circulation and formation of ideas are encoded in stylistic language.

In this paper, we propose a novel data collection approach that leverages cutting-edge LLMs along-side expert knowledge from philosophy scholars. This initiative marks the first effort to collect a comprehensive dataset InterIDEAS for philosophical intertextuality. This framework fosters cross-disciplinary efforts, bridging AI technology with philosophical scholarship to encourage practical insights and methodological advances. Our dataset comprises over 45,000 pages of texts by distin-guished philosophers spanning the years 1750 to 1950.

The main contributions are summarized as follows:

- We devise a schema that integrates LLMs' reading capacity and human expertise into the intertextual study of philosophy. The schema structures authentic philosophical writings in a manner that is organizable and analyzable by LLMs, demonstrating their potential in processing highly stylistic writing.

- We introduce InterIDEAS, a comprehensive open-source dataset [1] for achieving macroscopic views of the intellectual dynamics in philosophy. This dataset demonstrates a level of nuance and scale that ventures beyond traditional close-reading or bibliometric approaches.

- We perform preliminary experiments to showcase the utility of our dataset in both philosophy and AI research. The dataset not only uncovers the style, foundation, and intellectual tradition of modern philosophical inquiry, but also supplies an effective corpus for fine-tuning LLMs.

## 2 RELATED WORKS

**Existing Studies on Intertextuality**

Inquiry in intertextuality has been manually conducted by sociologists of philosophy like Randall Collins, who plotted network diagrams depicting philosophers' personal relationships, educational affiliations, and intellectual lineages according to his own extensive reading (Collins, 2009). However, the innately limited recollection, speed, and processing of human reading subject Collins' project to criticism like bias in text selection and interpretative methodologies.

The integration of NLP into knowledge extraction for text based data has markedly improved the capabilities of search systems in interpreting and processing human language and text, moving from simple keyword-based searches to complex analyses. There exist some computational explorations of intertextuality, including *Hyperhamlet* (a database gathering a corpus of references to Hamlet in literature, (Hohl Trillini & Quassdorf, 2010)), *Digital Dante* (a database mapping relations among writings by Dante and Ovid (Van Peteghem, 2020), and *EDHIPHY* (a database extracting Anglo-American philosophers' mentioning of each other in academic publications). In the first two examples, relations are drawn from a few texts to address very specific research interests. In the third case, while mentions are vital for macroscopic relational networks and indexical purposes, they

---

[1] https://anonymous.4open.science/r/InterIDEAS_data-9F56

cannot support more qualitative analysis; for the database only record the frequency of mentions, effacing their content and purposes.

**Advanced Automation for Humanities Studies**

The advent of deep learning, particularly through Transformer Vaswani (2017) based models such as BERT (Devlin et al., 2018), has led to a substantial paradigm shift in the area. These models excel in recognizing the nuances of language, greatly improving the accuracy of search results and accommodating more natural, conversational query inputs. Scholars have employed data-driven approaches and natural language processing (NLP) in studying dense writing, investigating topics like patterns in titles (Moretti, 2009) and abstracts (Ahlgren et al., 2015), authorial attribution (Peng & Hengartner, 2002), computational representation of arguments (Thagard, 2018), etc. However, traditional transformers face several limitations, such as restricted context understanding, poor reasoning capabilities, and limited knowledge integration, creating bottlenecks in humanities research that require deeper contextual analysis and cross-disciplinary insights.

Recent advances in LLM such as GPT-3, T0, Galactica and LLaMa (Sanh et al., 2021; Touvron et al., 2023; Taylor et al., 2022) have marked significant developments in NLP, in which GPT-4, the latest product, has notably enhanced capabilities in language understanding, generation, and reasoning. Multiple works have adopted LLMs for manufacturing textual datasets. The NORMDIAL dataset explores social norm adherence and violations in dialogue systems, leveraging LLMs to generate culturally contextual conversations, pushing the boundaries of cross-cultural language modeling (Li et al., 2023). In addition to dialogue, recent work on PoemSum (Mahbub et al., 2023) tests their ability to summarize poetry while retaining deeper figurative meanings. In the academic writing domain, the DOOLITTLE dataset, paired with reinforcement learning techniques, has shown promise in enhancing LLMs' capacity to generate formal, academic-level writing, showcasing potential improvements in GPT-4's stylistic and grammatical refinement abilities (Diao et al., 2023).

Although LLMs have proven effective in NLP dataset manufacturing and other general NLP tasks (Chang et al., 2024), their application in niche humanities areas, such as philosophy, is less examined. Thus, in this work, we propose a framework that integrates prompt tuning, retrieval-augmented generation (RAG), and HITL examination to generate answers for intertextuality-related questions on philosophical texts. Our dataset approaches intertextuality through semantic interpretation of full texts of authentic philosophical writings, moving beyond making comparisons at the word level and gathering statistics according to predetermined keywords, syntax, and formulated content. LLMs' effective comprehension of texts enables us to devise a descriptive and evaluative schema to collect copious references without effacing their content, function, and attitude reflected in detailed word and syntax choices.

## 3 CROSS-REFERENTIAL DATA COLLECTION

The collection of copious cross-referential entries through big data analysis and the careful literary interpretation of intertextuality through semantic details were once incompatible. Thankfully, the rapid advancements in LLMs have not only enabled the integration of both methods organically but also extended bibliometrics studies to humanities (Meyer et al., 2023; Zhong et al., 2023), where references varying in formats are often absent from the traditional bibliographies. Our goal is to allow LLMs to learn the patterns of philosophical texts and efficiently handle the cross-referential data from those texts through RAG and prompt engineering. Note that, the collected cross-referential data includes the references, ranging from casual mentioning, direct quotations, to extensive critiques, to other people, texts, and social and intellectual groups in political philosophy.

In this section, we outline the data collection workflow, focusing on how LLMs can learn from philosophical texts while preventing hallucination. We first process philosophical text into representations in appropriate lengths through Retrieval Augmented Generation (RAG) for enhancing context understanding. Next, we design prompts through multiple prompt engineering techniques to instruct LLMs to provide more accurate answers. Finally, through evaluation and attribution by human experts, the prompts will get modifications for improvement in a human-participant loop. Last but not least, for validating the effectiveness of the proposed LLM-based philosophy learning framework, we also evaluate the holistic quality of the manufactured datasets.

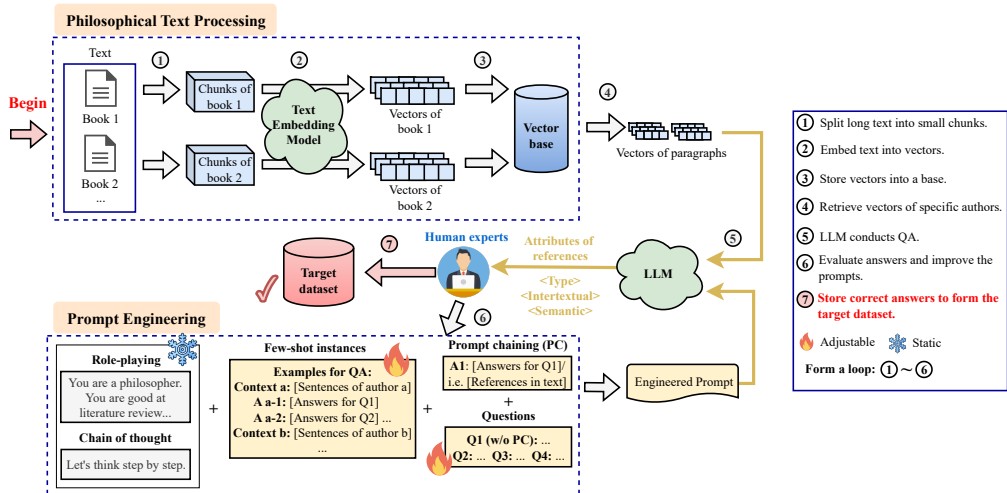

Figure 1: The entire workflow of the proposed data collection framework.

## 3.1 DATA COLLECTION WORKFLOW

Fig. 1 shows the workflow of our data collection approach, where the vector base is regarded as the information retrieval component of RAG, a technique for enhancing LLMs' knowledge by external information (Lewis et al., 2020), to augment the text generation of LLMs. To explain this figure, let us start with a book of a philosopher: 1) We first divide this book into text chunks (①), and embed them as representation vectors using text embedding models (②). Then, these vectors are stored in a vector base (③). 2) We initialize the prompts for Question-Answering (QA). Prompt engineering (White et al., 2023) is conducted on the questions about references in this book, to obtain a set of effective prompts. 3) When performing QA on this book, the representation vectors related to references are retrieved from the vector base (④), combined with the engineered prompts, and fed into an LLM to obtain the attributes of references, which are elaborated in Section 4.1. (⑤). Human experts in philosophy are requested to provide feedback and analysis on the answers to iteratively update and optimize the engineered prompts (⑥). Finally, high-quality responses from the LLM are paired correspondingly and stored in the database (⑦).

**a. Philosophical Text Processing**

For consistency of processing, texts in any form are transformed into PDF documents. Then, to fit into the LLM's context window, long pieces of text are split up into small, semantically congruent chunks. For reference QA, texts are split into paragraphs, since references often involve extensive analysis of details based on immediate context. When handling texts, we propose three parameters, including content type, intertextual function, and sentiment, to describe each reference, ensuring that these features are argumentatively critical to our selected oeuvre, and their evaluations can be satisfactorily performed by LLMs. Beside intertextual connections based on philosophers' overt engagement with each others' ideas, we are interested in their latent resonances and comparability. These can be loosely viewed as a type of intertextuality that an ideally erudite reader will construct for interpretative efficacy.

**b. Prompt Engineering**

To improve the quality and logic of the LLM's answers towards texts, as illustrated in Fig. 2, our framework employs a series of prompt engineering techniques, which are summarized below:

*1) Role-Playing (RP)*: We make the LLM play as a philosopher in the prompts, which is the Static Information in Fig. 2. From the perspective of a specific character, LLMs can generate more professional answers.

*2) Chain of Thought (CoT)* (Wei et al., 2022): In this framework, all questions are expanded with elaborate explanations and instructions. The principle is to break down complex problems into simpler steps with more detailed explanations to guide LLMs through a reasoning process. We define

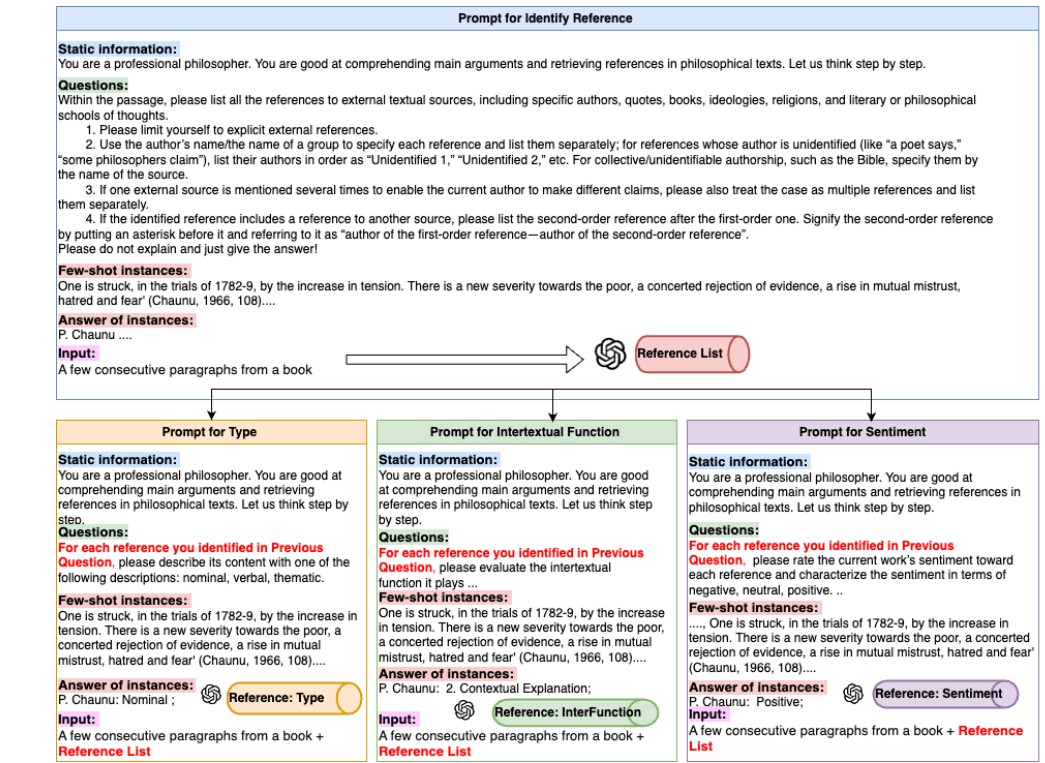

Figure 2: Prompting the LLM through few-shot examples to identify references, and evaluate their types, intertextual functions, and sentiments.

the problem by initially identifying the reference, the upper blue box of Fig. 2, followed by the assignment of three parallel tasks as the three lower-level boxes in Fig. 2.

*3) Few-Shot prompting (FS)* (Brown et al., 2020): This technique is adopted for reference QA. Multiple examples of context and corresponding references as well as other information are provided to help the model understand the questions as Few-shot Instances and Answer of Instances in Fig. 2.

*4) Prompt Chaining (PC)* (Wu et al., 2022): This technique is adopted for reference QA and semantic clustering. The answers from the LLM can be used as the input of the following prompts asking about any detail to guarantee the consistency of multi-step QA. Here, we use the input from the prompt to identify the reference to rest of the task.

We provide more prompt examples in Appendix A.

**c. Answer Evaluation and Prompt Improvement by Human Expert**

In addressing the intricacies of LLM outputs, we propose the addition of a dedicated phase, paralleling those on prompt and text processing, to delineate the role of expert intervention in refining these outputs. The goal of this phase is to iteratively enhance the prompt representation and address recurrent mistakes made by the LLM. Our approach hinges on the employment of seasoned experts, each over a decade of domain-specific experience, to manually review and correct the outputs of the LLM. Each time the LLM provides answers to a set of texts, human experts evaluate their accuracy and identify patterns in the errors. These identified patterns are then integrated into the respective question prompt as additional conditions. When the identified patterns of errors are difficult to express within a few words, the sentences will be added to few-shot instances as representative cases.

## 3.2 DATA QUALITY EVALUATION

To confirm the accuracy and showcase the efficacy of our approach in facilitating the comprehension of philosophical texts, this study is designed to assess and contrast the proficiency of our collection approach with that of human experts, humanities students, other students and LLM-only approaches

in identifying and extracting detailed information from philosophic materials (approximately 500 words each) sourced from 20th-century philosophical texts.

In our experiment, human experts comprise individuals who have obtained advanced degrees in fields such as literature or philosophy. The group of students with Bachelor's degrees in the humanities (BoH in Table 1) consists of individuals who have and only have obtained a Bachelor's degree in fields like literature or philosophy. The other student cohort includes native and non-native English speakers attending college to study the sciences, possessing a wide range of English language proficiency levels. For the purpose of this study, we recruited 5 human experts, 16 humanities students and 29 students of other backgrounds in both Australia and the United States, aiming to ensure a diverse and representative sample of participants for a comprehensive comparison of information extraction capabilities across different demographic groups. LLM-only approaches include ChatGPT3.5, ChatGPT3.5 with few-shot examples, ChatGPT4 and ChatGPT4 with few-shot examples. At the outset of the experiment, all participants received comprehensive instructions outlining the experimental requirements. They were then tasked with identifying and categorizing all references within a given paragraph in a strict timeframe of 20 minutes.

The performance of the approach is evaluated by its accuracy and recall in responding to each passage, and these results are juxtaposed with the outcomes from human participants. The evaluation of each response adheres to a uniform scoring system: Recall $= \frac{x}{y}$ and Accuracy $= \frac{x}{r}$, where $r$ is the total number of correct answers; $y$ is the total number of answers given by the participants and $x$ is the number of correct answers identified by the participants.

Table 1: Evaluation matrix. Bold numbers indicate the highest results from $P_1$-$P_6$ following human experts.

| | Accuracy | | | | | | Recall | | | | | |
|---|---|---|---|---|---|---|---|---|---|---|---|---|
| | $P_1$ | $P_2$ | $P_3$ | $P_4$ | $P_5$ | $P_6$ | $P_1$ | $P_2$ | $P_3$ | $P_4$ | $P_5$ | $P_6$ |
| Human Experts | 1 | 1 | 1 | 0.92 | 0.89 | 0.98 | 1 | 1 | 1 | 1 | 0.93 | 1 |
| Student/w.BoH | 0.97 | 0.75 | 0.63 | 0.75 | 0.75 | 0.64 | 0.85 | 0.74 | 0.79 | 0.66 | 0.56 | 0.71 |
| Other Students | 0.75 | 0.6 | 0.68 | 0.47 | 0.44 | 0.75 | 0.69 | 0.62 | 0.68 | 0.47 | 0.25 | 0.60 |
| ChatGPT3.5 | 0.46 | 0.58 | 0.66 | 0.71 | 0.67 | 0.63 | 0.54 | 0.61 | 0.53 | 0.47 | 0.25 | 0.43 |
| ChatGPT3.5/w.FS | 0.75 | 0.55 | 0.71 | 0.63 | 0.8 | 0.75 | 0.69 | 0.55 | 0.53 | 0.41 | 0.50 | 0.60 |
| ChatGPT4/w.FS | 0.75 | 0.64 | 0.6 | 0.65 | **0.83** | 0.74 | 0.69 | 0.64 | 0.6 | 0.77 | 0.63 | 0.66 |
| **Ours** | **0.85** | **0.91** | **0.8** | **0.74** | 0.75 | **0.84** | **0.85** | **0.91** | **0.8** | **0.81** | **0.75** | **0.88** |

Table 1 shows the experimental results. Rows labeled Student/w.BoH, ChatGPT3.5/w.FS, and ChatGPT4/w.FS in the table correspond to the experimental results for the baselines: students with a Bachelor's degree in Humanities, ChatGPT-3.5 using few-shot examples, and ChatGPT-4 using few-shot examples, respectively. Columns $P_1$ through $P_6$ in the table detail the accuracy and recall results for all baselines and our method, as applied to experiments on philosophical materials 1 through 6. Human experts outperform others, with amateurs struggling to grasp complex texts. Our approach ranks just below experts, excelling in accuracy and recall measures the model's correct responses, indicating its precision. Recall assesses its ability to identify all relevant answers. Higher accuracy but lower recall suggests the model may miss some correct answers, whereas higher recall but lower accuracy indicates it identifies many answers, but not all are correct. Although human experts achieve superior extraction outcomes compared to our method, the resource of human experts is very limited and costly. Thus, the experimental results verify that our method is effective, efficient, and economic, particularly in processing large-scale philosophical texts.

## 4 INTERIDEAS DATASET OVERVIEW

In this study, we restrict our focus to books originally written or have been translated in English. To date, we have analyzed over 45,000 pages of modern philosophy available in English. Our dataset has amassed over 15,000 cross-referential data pairs, encompassing more than 3,150 philosophers and philosophical schools, covering the majority of both during this period. Our periodization corresponds to the so-called "modern period" in the humanities. Despite its lack of pinpointable timeline, the usual consensus is that the modern period is loosely bound by the beginning of the Industrial Revolution (circa 1760) and the end of WWII (1945). We slightly extended the timeline to address the time lag between historical events and their intellectual stimuli and reactions. In selecting texts, we balanced coverage with representativeness. We incorporated authors and texts into the dataset according to

three objectives: 1) Covering prominent thinkers; 2) Featuring different geographical locations for intellectual debates, including traditional cultural centers like France, emerging intellectual hubs at that time like the U.S., and marginalized places like India); 3) Presenting writings from authors of different occupations, including academics, journalists, political activists, novelists, and literary critics.

## 4.1 METADATA FORMAT

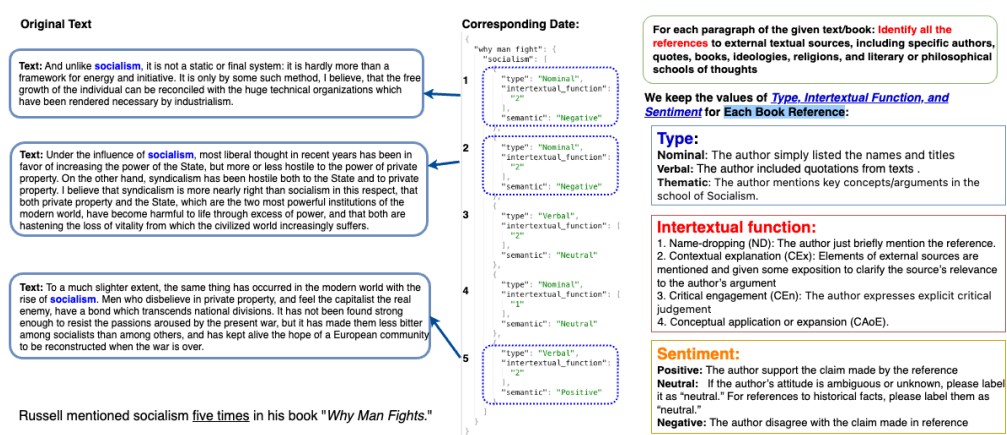

Figure 3: Metadata Format and Description.

Empirically speaking, most discussions of external materials in philosophy fall into the following categories: ideas or activities of specific agents or groups. Therefore, we delineate intertextuality as references to other discourses, including books, ideologies, religions, historical events, and words and deeds of other people. With our deliberately loose definition guiding LLMs to extract references of diverse nature—ranging from published texts to anecdotes, from specific individuals to vague social groups—the dataset reflects different philosophical, political, historical, and personal components that jointly contribute to the vibrancy of modern philosophy.

As shown in Fig. 3, we present a metadata schema specifically designed for the analysis of intertextual references within humanities writing. The schema facilitates the categorization and detailed examination of references, and their content type, intertextual functions, and sentiment.

The provided dataset is an organized compilation of bibliographic entries related to philosophy books, encompassing detailed attributes for each book. These attributes include the *Book Title*, the *Reference Name*, Linked directly to each reference is the *Content Type*, which provides detailed information sorted into the nominal, the verbal, and the thematic. This entity captures the essence of each reference through the identification of specified names and titles, presenting quotations from other texts, and giving brief summaries for loose, unspecified discussion of external references, respectively. Each reference is also associated with an *Intertextual Function*, which describes the role the reference plays in the text—ranging from name-dropping (ND) and contextual explanation (CEx) to critical engagement (CEn) and conceptual application or expansion (CAoE). This classification helps us understand the extent of interaction between the current work and the referred content. Furthermore, the *Sentiment* assesses the current author's sentiment towards each reference, which is categorized as negative, neutral, or positive. This evaluation is crucial for discerning the author's perspective and the reference's intended effect on readers' understanding. The relationships among these values are structured to ensure an one-to-one correspondence between a reference and its content type, intertextual function, and sentiment.

Based on our dataset, Nominal references are the most common, constituting 67.9% of the data, followed by thematic and verbal references. In sentiment analysis, neutral sentiments predominate at 77.4%, with positive and negative sentiments at 13.1% and 9.5% respectively. For intertextual functions, name-dropping is most frequent, making up 52% of the instances, whereas critical engagement and contextual explanation are also significant, and conceptual application or expansion is relatively rare. These statistics illustrate the dominance of nominal referencing and neutral sentiments in the dataset, with name-dropping being the primary intertextual function. Meanwhile, authors'

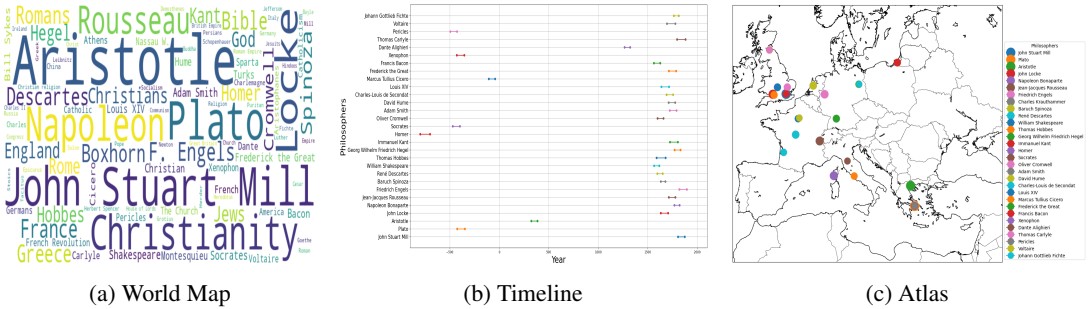

(a) World Map      (b) Timeline      (c) Atlas

Figure 4: Philosophical References: A Temporal and Geographical Overview

attitudes are crucial in determining the depth of their engagement with others' ideas and actions, as shown in Table 2. Negative attitudes are often suggested by explicit criticism. People, events, or works that the current authors feel impartial about are usually cursorily discussed. Our general statistics of our dataset also uncover features of modern philosophical writings. First, the dominance of neutral and positive sentiments show that the field is largely organized by amicability. Second, the distribution of sentiments across intertextual function suggests that in constructing philosophical arguments, philosophers generally adapt the style of discussion (recorded as "function" in the dataset) rather than the choice of materials ("type") to reflect their perspectives on individuals, schools of thought, and events ("sentiment").

Table 2: Distribution of sentiments across intertextual functions.

| Intertextual Function | Negative | Neutral | Positive |
|---|---|---|---|
| Name-dropping | 514 | 6537 | 778 |
| Contextual Explanation | 284 | 2626 | 657 |
| Critical Engagement | 620 | 2361 | 394 |
| Conceptual Application or Expansion | 12 | 119 | 145 |

## 5 APPLICATIONS OF INTERIDEAS IN PHILOSOPHY AND LLMS

### 5.1 ANALYSIS IN INTERIDEAS FOR PHILOSOPHY :

Our dataset facilitates diachronical and synchronical analyses of philosophy. We extract the 50 most frequent references appeared in at least 3 texts. The word map 4a confirms the interdisciplinary nature of philosophy . Besides acclaimed philosophers and philosophical schools, we find religions (e.g., "Christianity," "God," "Buddha," and "The Bible") and political events and entities (e.g., "Roman Empire," "British Empire," and "French Revolution") constitutive to philosophical discussion.

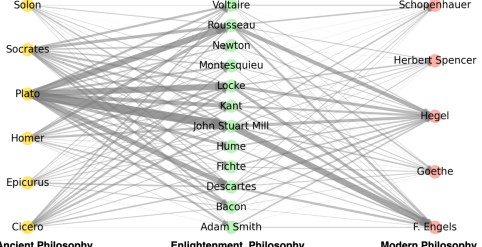

Figure 5: Flow of Thought: A Graph of Referenced Philosophers from Ancient to Modern Era

We extract all the individuals from these common references, plotting their life-span and major location of intellectual activity on a timeline 4b and a map 4c respectively. These visual representations show that modern philosophers regard ancient, enlightenment, and contemporaneous philosophy in the Mediterranean region and the English Channel region as their shared base of intellectual inquiry.

Upon this foundation, we connect writers of antiquity, the Enlightenment era, and the modern era by a mapping network 5 which tracks the flow of ideas. The network shows, for example, how likely a modern philosopher who has referred to Solon would also be influenced by Voltaire and moreover by Schopenhauer. Our chart suggests that two important intellectual tradition for modern philosophy are Plato-Rousseau-Hegel and Plato-John Stuart Mill-Engels.

## 5.2 SENTIMENT CLASSIFICATION ENHANCEMENT FOR LANGUAGE MODELS

To demonstrate the potential function of the proposed dataset in AI tasks, we create 2,236 reference-attitude pairs from our dataset. Each pair comprises a sentence from an authentic philosophical text and its author's assessed attitudes towards the referenced content. These pairs are divided into training (70%), validation (20%), and test sets (10%), where in the test set, samples with label "Negative", "Neutral", and "Positive" are 142, 53, and 33, respectively.

For validation, we consider not only LLMs but also pre-trained language models (PLMs) in our experiment. PLMs focus on pre-training to generate general language representations for downstream tasks, while LLMs primarily focus on natural language generation and typically involve larger model scales. Since both models can be fine-tuned to adapt to downstream tasks, we select five popular PLMs and four outstanding LLMs for fine-tuning. The five PLMs can be categorized into three types: 1) BERT-based: BERT (Devlin et al., 2018), ALBERT (Lan et al., 2019; Schuster et al., 2021), and BERTweet (Nguyen et al., 2020); 2) RoBERTa (Liu et al., 2019; Barbieri et al., 2020); 3) XLNet (Yang et al., 2019). On the other hand, the four LLMs can be classified into three types too: 1) Llama-based: Llama 2-7B (Touvron et al., 2023) and Llama 3-8B (Touvron et al., 2023; Wang et al.; 2024); 2) Mistral-7B (Jiang et al., 2023; Dong et al., 2023; Xiong et al., 2024); 3) GPT-2 (Radford et al., 2019). Additionally, we also study GPT-4o (Achiam et al., 2023), which is the most state-of-the-art (SOTA) LLM, to do direct inference without any extra training. Furthermore, we randomly choose 5 samples of each label from the training set as the few-shot instances for GPT-4o. The performance of all PLMs and LLMs pre-trained for text/sequence classification is compared before and after fine-tuning on our reference-attitude dataset for 100 epochs.

Evaluation metrics include accuracy, macro F1 score, macro precision, and macro recall, to calculate more reasonable results of the imbalanced test set. Additionally, the size of each model, the proportion of fine-tuned parameters, and the time cost for fine-tuning are recorded in Table 3. For more clear demonstration, the confusion matrices of each model are shown and analyzed in Appendix J.

Table 3: Popular open-source PLMs and LLMs for sentiment classification on the proposed dataset w./w.o. fine-tuning, or few-shot learning for GPT-4.

| Model | Before fine-tuning/few-shot | | | | After fine-tuning/few-shot | | | | Computational cost | | |
|---|---|---|---|---|---|---|---|---|---|---|---|
| | Acc. | $F1$ | Pre. | Rec. | Acc. | $F1$ | Pre. | Rec. | Param. | FT % | Sec. |
| BERT | 16.67 | 14.24 | 28.36 | 30.26 | 63.32 | 39.01 | 51.59 | 39.69 | 0.11B | 1.21% | 69 |
| ALBERT | 14.91 | 9.72 | 16.00 | 33.96 | 60.96 | 25.25 | 20.59 | 32.63 | 0.05B | 0.24% | 32 |
| BERTweet | 28.51 | 22.28 | 36.44 | 34.94 | 60.96 | 34.23 | 37.48 | 36.57 | 0.13B | 0.98% | 61 |
| RoBERTa | 23.25 | 12.57 | 7.75 | 33.33 | 63.16 | 45.68 | 50.80 | 44.76 | 0.12B | 2.00% | 222 |
| XLNet | 28.07 | 24.73 | 37.81 | 38.19 | 49.56 | 35.48 | 35.45 | 35.54 | 0.12B | 0.62% | 245 |
| **Average** | **22.28** | **16.67** | **25.27** | **34.14** | **59.59** | **35.93** | **39.18** | **37.84** | - | - | - |
| Llama 2 | 26.75 | 25.39 | 35.52 | 29.17 | 62.28 | 53.17 | 54.03 | 52.49 | 6.54B | 0.50% | 677 |
| Llama 3 | 27.63 | 27.79 | 40.82 | 39.77 | 67.54 | 62.61 | 61.02 | 65.45 | 7.51B | 0.52% | 747 |
| Mistral | 25.88 | 25.59 | 32.68 | 36.81 | 50.44 | 45.20 | 45.30 | 49.98 | 7.11B | 0.94% | 859 |
| GPT-2 | 27.19 | 27.72 | 41.90 | 39.08 | 53.95 | 48.42 | 47.41 | 51.11 | 0.38B | 0.88% | 175 |
| **Average** | **26.86** | **26.62** | **37.73** | **36.21** | **58.55** | **52.35** | **51.94** | **54.76** | - | - | - |
| GPT-4 | 24.56 | 21.03 | 34.91 | 33.58 | 42.54 | 40.79 | 51.05 | 47.47 | - | - | - |

In Table 3, the average performance improvements before and after fine-tuning are noteworthy. The average accuracy of PLMs and LLMs increased from 22.28% and 26.86% to 59.59% and 58.55%, and the average F1 score improved from 16.67% and 26.62% to 35.93% and 52.35%, respectively. This demonstrates that our provided philosophical corpus exhibits significant potential for fine-tuning across various models. Overall, the accuracy of PLMs is generally slightly higher than that of LLMs, but the F1 scores are noticeably lower. This could be attributed to the fact that PLMs have significantly fewer parameters than LLMs, coupled with the presence of data imbalance in the training set (with more negative samples). As a result, overfitting during fine-tuning PLMs might have occurred, causing the outputs to be heavily biased towards the negative class. Besides, PLMs consume less computational resources compared to LLMs. This indicates that PLMs, while less resource-intensive, may struggle with achieving balanced performance across different classes in the context of imbalanced datasets, particularly in complex tasks like sentiment analysis of philosophical texts. Additionally, the results from GPT-4 show that even simple few-shot learning markedly improves

output quality. This validates the representational quality of our dataset samples. In conclusion, our corpus positively contributes to helping language models better understand the philosophical context.

Besides, we present confusion matrices of Llama 3 w./w.o. fine-tuning and GPT-4o w./w.o. few-shot learning adopted for sentiment classification in Fig. 6. Before fine-tuning or few-shot learning, all models tend to favor one class and do not consistently choose the Negative class, despite its abundance in the test set. This suggests that advanced language models often experience mode collapse in philosophical sentiment classification. After fine-tuning, models show a marked preference for negative sentiment, indicating improved performance through fine-tuning.

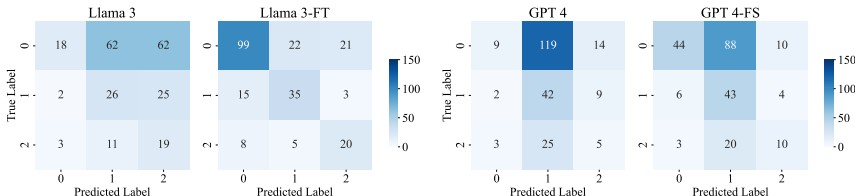

Figure 6: Confusion matrices of Llama 3 w./w.o. fine-tuning and GPT-4o w./w.o. few-shot learning adopted for sentiment classification.

## 6 LIMITATIONS

Limitations of using LLMs for processing philosophical texts found in our work are summarized as follows: *1) Semantic dissection*: When multiple references are listed in paralleling grammatical structures, the LLM may categorize them into different functions, even though they assume identical rhetorical roles. Through manual review, representative sentences are integrated into few-shot instances, and some constraints are imposed in the questions, effectively mitigating this issue. *2) Literal-mindedness*: The LLM struggles in literary expressions with complex emotions, such as rhetorical questions and irony. This aspect has seen some improvement through the addition of few-shot instances. *3) Stereotyping*: Faced with specific input information, such as "Hitler," the LLM tends to respond based on its built-in stereotypes with "negative" disregarding the author's potentially "neutral" or "positive" stance.

Limitations of our dataset are summarized as follows: *1) Style*: The dataset excludes symbol- and aphorism-based texts, which require the designing of a completely different approach to parse, collect, and analyze their intertextuality. Since symbols tend to be heavily featured in philosophical subfields like logic and philosophy of language, and since certain philosophers like Wittgenstein have a predilection for aphorisms, our dataset can potentially exclude a few topics and writers. *2) Language*: Our current approach is limited to texts that are written in or have been translated into English. This limitation can raise concerns of Eurocentrism. To address these problems, we hope to extend the approach to other styles and languages in the future by recruiting philosophical researchers with different research and language expertise.

## 7 CONCLUSION

In this paper, we introduce InterIDEAS, the first dataset for extracting and evaluating philosophical intertextuality. Enhanced by both LLMs and philosophical expertise, this dataset provides a robust foundation for exploring intellectual structures and dynamics through references. We propose a systematic methodology to categorize, analyze, and interpret complex relationships within and beyond philosophy. InterIDEAS elucidates the intricate ways in which different discourses influence each other, uncovering latent patterns in philosophy that offer insights to both philosophical studies and AI research.

REPRODUCIBILITY STATEMENT

In our commitment to reproducibility, this paper introduces a novel dataset that is fully documented and publicly available. We provide a comprehensive data description and accessible source code, enabling other researchers to implement our dataset across all test cases.

The paper and supplementary materials contain detailed documentation of the dataset, outlining the data collection process, preprocessing steps, and the parameters used during experiments. This documentation ensures that researchers can replicate the dataset creation and test it under the same conditions reported in our study.

Furthermore, we have made both the dataset and the source code available for downloading through an anonymous link. This code includes scripts for data handling, model implementation, and parameter setting, which correspond to all test cases presented in this paper.

By making these resources available, we aim to provide a transparent and accessible framework that allows researchers to reproduce and build upon our work, fostering further scientific exploration and validation.

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

## A  PROMPTS FOR REFERENCES

**Prompt for Reference 1**

**Static information:**
You are a professional philosopher. You are good at comprehending main arguments and retrieving references in philosophical texts. Let us think step by step.

**Question:**
Within the passage, please list all the references to external textual sources, including specific authors, quotes, books, ideologies, religions, and literary or philosophical schools of thoughts.
1. Please limit yourself to explicit external references.
2. Use the author's name/the name of a group to specify each reference and list them separately; for references whose author is unidentified (like "a poet says," "some philosophers claim"), list their authors in order as "Unidentified 1," "Unidentified 2," etc. For collective/unidentifiable authorship, such as the Bible, specify them by the name of the source.
3. If one external source is mentioned several times to enable the current author to make different claims, please also treat the case as multiple references and list them separately.
4. If the identified reference includes a reference to another source, please list the second-order reference after the first-order one. Signify the second-order reference by putting an asterisk before it and referring to it as "author of the first-order reference—author of the second-order reference".
Please do not explain and just give the answer!

**Few-shot instances:**

**Context:**
One is struck, in the trials of 1782-9, by the increase in tension. There is a new severity towards the poor, a concerted rejection of evidence, a rise in mutual mistrust, hatred and fear' (Chaunu, 1966, 108).
...
Homage is paid to the 'great reformers' - Beccaria, Servan, Dupaty, Lacretelle, Duport, Pastoret, Target, Bergasse, the compilers of the Cahiers, or petitions, and the Constituent Assembly - for having imposed this leniency on a legal machinery and on 'classical' theoreticians who, at the end of the eighteenth century, were still rejecting it with well-formulated arguments.
...
What is this nationalist political theory about? ... This is opposed to imperialism, which seeks to bring peace and prosperity to the world by uniting mankind, as much as possible, under a single political regime. ... At that time, the struggle against Communism ended, and the minds of Western leaders became preoccupied with two great imperialist projects ...

**Answers of instances:**
P. Chaunu; Beccaria, Servan, Dupaty, Lacretelle, Duport, Pastoret, Target, Bergasse; Imperialism; Communism; ...

**Prompt for Reference 2**

**Static information:**
You are a professional philosopher. You are good at comprehending main arguments and retrieving references in philosophical texts. Let us think step by step.

**Question:**
For each reference you identified in question 1, please describe its content with one or more of the following descriptions:
1. Nominal, meaning those references that explicitly mention names of other authors, books, collections of works, and other schools of thought in the main text; for nominal references, signal their content by exact names used in the passage. Specification of authors or sources in citational practice does not count as nominal. If there are multiple nominal references, separate them by colons.
2. Verbal, meaning direct quotation of phrases and sentences from other sources; for verbal references, signal their content by abbreviated versions of the quotes that only keep the first and the last two words of the quote, with ellipses in between. If there are multiple verbal references, separate them by colons.
3. Thematic, meaning references to others' claims, ideas, and motifs not through direct quotes but through paraphrases; for thematic references, please signify their content by a summary in one or two philosophical terms. If there are multiple thematic references, separate them by colons.
If there is no reference to others' claims in a category, please give NA.
If one external source is mentioned several times to enable the current author to make different claims, please also treat the case as multiple references and list them separately.
Lastly, formulate your answer in this way:
Referred item: nominal (content of the nominal references); verbal (content of the verbal references); 3. thematic (content of the thematic references)
Please do not explain and just give the answer!

**Few-shot instances:**
In these few shot examples, we covered all the cases. When you run the prompt, please choose the most applicable one for each reference. You don't need to identify all functions within a passage.
These are examples for your answer:

**Context:**
The same as the context in Fig. 7.

**Answers of instances:**
P. Chaunu: Nominal (P. Chaunu); Verbal ("a constant... for security"); Thematic (crime; economic pressure);
Beccaria, Servan, Dupaty, Lacretelle, Duport, Pastoret, Target, Bergasse: Nominal (Beccaria, Servan, Dupaty, Lacretelle, Duport, Pastoret, Target, Bergasse, Cahiers);
Imperialism: Thematic (Alternative to nationalism);
Communism: Thematic (the Cold War);
...

Figure 7: The engineered prompt for the 1st question for references.

Figure 8: The engineered prompt for the 2nd question for references.

**Prompt for Reference 3**

**Static information:**
You are a professional philosopher. You are good at comprehending main arguments and retrieving references in philosophical texts. Let us think step by step.

**Question:**
For each reference identified in question 1, please evaluate the intertextual function it plays by the closet descriptions below. Classify the references by "Name-Dropping," "Contextual Explanation," "Critical Engagement," or "Conceptual Application or Expansion";

1. Name-Dropping: This category is for when the current work merely mentions the names of authors, works, or concepts as representative cases of a phenomenon or an argument, without detailed explanations that exceed one sentence. In particular, if there is a list of names whose individual significance is not discussed, please label them as "Name-Dropping." Other markers for this category include mentioning in passing like "c.f.," "for details, please see...," etc.

2. Contextual Explanation: Elements of external sources are mentioned and given some exposition to clarify the source's relevance to the author's argument. These references add depth to the discussion but are presented without the author's personal judgment of the reference as right or wrong. Examples include references to factual evidence in support of the argument, references that intend to exemplify the author's arguments, etc.

3. Critical Engagement: In this category, the current work actively engages with external sources by offering detailed analysis (at least one sentence of analysis for each reference) and value judgements. The author's subjective attitudes are evident as they express their agreements or disagreements with the ideas presented in these references.

4. Conceptual Application or Expansion: References that fall into this category are not only explained but are also used as a springboard for further development of the current work. The current work distills keywords or arguments from the reference and expands upon them, possibly transforming them or integrating them into a new framework. Examples include a problematic concept that is adjusted and employed in further discussion; a methodology from other sources is adopted by the current author, etc.

If one external source is mentioned several times to enable the current author to make different claims, please also treat the case as multiple references and list them separately.
Please do not explain and just give the answer!

**Few-shot instances:**
In these few shot examples, we covered all the cases. When you run the prompt, please choose the most applicable one for each reference. You don't need to identify all functions within a passage.
These are examples for your answer:

**Context:**
The same as the context in Fig. 7.

**Answers of instances:**
P. Chaunu: 2. Contextual Explanation;
Beccaria, Servan, Dupaty, Lacretelle, Duport, Pastoret, Target, Bergasse: 1.Name-dropping;
Imperialism: 2. Contextual Explanation;
Communism: 1.Name-dropping;
...

Figure 9: The engineered prompt for the 3rd question for references.

**Prompt for Reference 4**

**Static information:**
You are a professional philosopher. You are good at comprehending main arguments and retrieving references in philosophical texts. Let us think step by step.

**Question:**
Please rate the current work's sentiment toward each reference identified in question 1, and characterize the sentiment in terms of negative, neutral, positive. If the author's attitude is ambiguous or unknown, please label it as "neutral." For references to historical facts, please label them as "neutral." For second-order references, please assess the author's sentiment to the second-order reference, not the sentiment of the first-order reference to the second-order reference. Please base your judgment only on the provided passage.

If one external source is mentioned several times to enable the current author to make different claims, please also treat the case as multiple references and list them separately.

**Few-shot instances:**
In these few shot examples, we gave examples for all sentiments. In your application, please select the most appropriate sentiment. You don't have to find traces of all sentiments within a given passage.
These are examples for your answer:

**Context:**
The same as the context in Fig. 7.

**Answers of instances:**
P. Chaunu: Positive;
Beccaria, Servan, Dupaty, Lacretelle, Duport, Pastoret, Target, Bergasse: Neutral;
Imperialism: Neutral;
Communism: Neutral;
...

Figure 10: The engineered prompt for the 4th question for references.

## B  LICENSING

All the data we currently open to public are originating from Project Gutenberg `https://gutenberg.org/about/`. Project Gutenberg eBooks may be freely used in the United States because most are not protected by U.S. copyright law. They may not be free of copyright in other countries. Readers outside of the United States must check the copyright terms of their countries before accessing, downloading or redistributing eBooks. We also have a number of copyrighted titles, for which the copyright holder has given permission for unlimited non-commercial worldwide use. For Project Gutenberg, no permission is needed for non-commercial use. So, for example, you can freely redistribute any eBook, anywhere, any time, with or without the "Project Gutenberg" trademark included. The "Small Print" has more details. Note that if you are not in the US, you must confirm yourself whether an item is free to redistribute where you are.

The copyright status of philosophy books can vary significantly depending on several factors, such as the date of the author's death and the specific laws of the country in which the book was published. Here are some general guidelines: In most countries, works enter the public domain 70 years after the death of the author. If the author of a philosophy book died more than 70 years ago, it is likely that their works are now in the public domain. Besides, some philosophy books, especially classic texts, may be in the public domain, but newer editions (which might include modern commentary, translations, or annotations) can still be protected by copyright. Copyright laws can vary from one country to another. For example, some countries have extensions for certain types of works or authors.

For the remaining unpublished data, we are actively working on verifying the copyright status and obtaining the necessary permissions. We will continue to update our dataset as soon as we confirm the copyright status of each book and secure the appropriate permissions.

## C  ACCURACY FOR WHOLE DATASET

Given the lack of available tools other than human expertise for verifying the accuracy of the resulting dataset, and considering the impracticality of human experts reviewing all responses due to the extensive volume of material, we have adopted a strategy of randomly selecting 5 text chunks per 100 for manual verification. Additionally, we plan to make this dataset accessible for future research use and will provide an interface allowing users to identify errors and update the dataset accordingly. Based on the random sample and check, ChatGPT showed remarkable precision in recognizing 98.11% of references to external sources across all books. Additionally, it was able to accurately depict 93% of the content from these identified references. As of the current date, language learning models (LLMs) have achieved a 75.7% success rate in identifying intertextual functions and an 86.4% success rate in sentiment analysis.

At this stage, our goal is to confirm that the performance of the LLM is stable across texts. Verifying its performance on a random 5% pages for each book we processed is sufficient to reflect its overall performance. Meanwhile, 5% of 45000 pages is 2250 pages. Each of our human experts spent on average 10 minutes reading a page, processing 15- 20 pages per day. 5% is already a taxing workload.

## D  HUMAN READING CAPABILITY EXPERIMENT

### D.1  INSTRUCTIONS

**Objective:** The aim of this experiment is to assess the intertextual reading ability of individuals at various levels of proficiency. Participants will be asked to read texts of differing complexity and respond to the listed questions. we focus on assessing LLM performance against general human performance, not just versus experts. We include both expert and non-expert readers of philosophical texts. The results show that LLMs perform better than nonprofessionals, though they fall short of expert levels. This suggests that our dataset can expand experts' analytic scope and improve nonprofessionals' understanding of textual details. It alsp implies that the task requires specialized knowledge or skills that are beyond the capacity of general participants and highlights the effectiveness of the LLM in handling complex scenarios where typical human capabilities are insufficient. Such findings might be essential for understanding the limits of human performance in specific contexts and the potential areas where advanced models like LLMs can be particularly beneficial.

**Participant Requirements:**

- Age: 20-80
- Language Proficiency: Participants must be college students or individuals with higher education, residing in an English-speaking country.

**Materials Provided**

- A series of texts at varying levels of difficulty.
- A questionnaire for each text to assess intertextual reading ability.

### D.1.1  PROCEDURE

**Introduction:** Participants will receive an overview of the experiment, including its purpose and what will be required of them.

**Consent:** Participants must read and sign a consent form agreeing to partake in the experiment and acknowledging the confidentiality and use of their data.

**Pre-Test Survey:** A short survey to gather participant background information relevant to the study, such as age, education level, and reading habits.

**Pre-Reading:** Participants will give 15 minutes to read the instruction for questions

**Reading Task:** Participants will be given one or two texts, Each text should be read in a quiet environment without distractions. Participants are advised to read at their natural pace.

**Comprehension Assessment:** After reading each text, participants will answer a set of questions. The questions may be multiple choice, short answer, or a mix of both.

**Breaks:** Participants are allowed to take short breaks between texts if needed.

**Post-Reading Survey:** After completing all the readings, participants will fill out a survey capturing their experience, challenges faced, and any feedback on the texts.

**Debriefing:** Participants will be provided with a summary of the experiment and its objectives. Any questions or concerns from participants will be addressed.

### D.1.2 ETHICS AND CONFIDENTIALITY

All participant information will be kept confidential. Participants have the right to withdraw from the study at any point without any negative consequences.

### D.1.3 CONTACT INFORMATION

Provide contact details for participants to reach out if they have any questions or concerns before, during, or after the experiment. Thank you for your participation and valuable contribution to this research!

### D.1.4 COMPENSATION

Each participant is provided with a $15 coupon for the school coffee shop.

### D.2 QUESTIONS

### D.2.1 Q1 FOR REFERENCE IDENTIFICATION

Within the passage, please list all the references to external textual sources, including specific authors, quotes, books, ideologies, religions, and literary or philosophical schools of thoughts. Use the author's name/the name of a group to specify each reference; for references whose author is unidentified (like "a poet says," "some philosophers claim"), list their authors in order as "Unidentified 1," "Unidentified 2," etc. For collective/unidentifiable authorship, such as the Bible, specify them by the name of the source.

### D.2.2 Q2 FOR CONTENT TYPE

For each reference you identified, please describe its content with one or more of the following descriptions: 1. Nominal, meaning those references that explicitly mention names of other authors, books, collections of works, and other schools of thought in the main text; for nominal references, signal their content by exact names used in the passage. If there are multiple nominal references, separate them by colons. E.g., Marx: nominal (Marx; The Communist Manifesto) 2. Verbal, meaning direct quotation of phrases and sentences from other sources; for verbal references, signal their content by abbreviated versions of the quotes that only keep the first and the last two words of the quote, with ellipses in between. If there are multiple verbal references, separate them by colons. E.g., Marx: verbal ("the history...class struggles") 3. Thematic, meaning references to others' claims, ideas, and motifs not through direct quotes but through paraphrases; for thematic references, please signify their content by a summary in one or two philosophical terms. If there are multiple thematic references, separate them by colons. E.g., Marx: thematic (child labor)

### D.2.3 Q3 FOR INTERTEXTUAL FUNCTION

For each reference identified in prompt 1, please evaluate the intertextual function it plays by the closet descriptions below. Classify the references by "Name-Dropping," "Contextual Explanation," "Critical Engagement," or "Conceptual Application or Expansion." 1. Name-Dropping: This category is for when the current work merely mentions the names of authors, works, or concepts as representative cases of a phenomenon or an argument, without detailed explanations. 2. Contextual Explanation: Elements of external sources are mentioned and given some exposition to clarify the source's relevance to the author's argument. These references add depth to the discussion but are presented without the

author's personal judgment of the reference as right or wrong. Examples include references to factual evidence in support of the argument, references that intend to exemplify the author's arguments, etc. 3. Critical Engagement: In this category, the current work actively engages with external sources by offering detailed analysis (at least one sentence of analysis for each reference) and value judgements. The author's subjective attitudes are evident as they express their agreements or disagreements with the ideas presented in the reference. 4. Conceptual Application or Expansion: References that fall into this category are not only explained but are also used as a springboard for further development of the current work.

### D.2.4 Q4 FOR SEMANTIC

Please rate the current author's sentiment toward each reference identified in prompt 1, and characterize the sentiment in terms of strongly negative, negative, neutral, positive, strongly positive. If the author's attitude is ambiguous or unknown, please label it as "neutral". For references to historical facts, please label them as "neutral". Organize your final answer as: Marx Nominal (Marx; The Communist Manifesto); Verbal ("the history…class struggles"); Thematic (child labor) 3. Critical Engagement Positive

## E   STATIC ANALYSIS FOR THE DATA QUALITY EVALUATION

### E.1   ACCURACY

Table 4: Summary of accuracy results with statistical analysis.

| Group | Scores | | | | | | Average | Std. Dev. | P-value |
|---|---|---|---|---|---|---|---|---|---|
| Human Experts | 1 | 1 | 1 | 0.92 | 0.89 | 0.98 | 0.965 | 0.044 | 0.039 |
| Student/w.BoH | 0.97 | 0.75 | 0.63 | 0.75 | 0.75 | 0.64 | 0.748 | 0.112 | 0.110 |
| Other Students | 0.75 | 0.6 | 0.68 | 0.47 | 0.44 | 0.75 | 0.615 | 0.124 | 0.258 |
| GPT3.5 | 0.46 | 0.58 | 0.66 | 0.71 | 0.67 | 0.63 | 0.618 | 0.081 | 0.382 |
| GPT3.5/w.FS | 0.75 | 0.55 | 0.71 | 0.63 | 0.8 | 0.75 | 0.698 | 0.084 | 0.523 |
| GPT4/w.FS | 0.75 | 0.64 | 0.6 | 0.65 | 0.83 | 0.74 | 0.702 | 0.079 | 0.659 |
| Ours | 0.85 | 0.91 | 0.8 | 0.74 | 0.75 | 0.84 | 0.815 | 0.059 | 0.722 |

Human Experts have the highest consistency with an average score of 0.965 and a standard deviation of 0.044. Their performance distribution may not be normal (p-value = 0.039). Student with BoH shows moderate variability with an average of 0.748 and a standard deviation of 0.112, with performance deemed normally distributed (p-value = 0.110). Other Students have the most variability with an average of 0.615 and a standard deviation of 0.124, and normal distribution (p-value = 0.258). GPT3.5 and GPT3.5 with FS score averages of 0.618 and 0.698, respectively, both with normal performance distributions (p-values > 0.380). GPT4 with FS and GPT4 with FPEh show consistent high performance with averages of 0.702 and 0.815, respectively, and low variability (SD < 0.08), with normal distribution (p-values > 0.650).

### E.2   RECALL

Table 5: Summary of recall results with statistical analysis.

| Group | Scores | | | | | | Average | Std. Dev. | P-value |
|---|---|---|---|---|---|---|---|---|---|
| Human Experts | 1 | 1 | 1 | 1 | 0.93 | 1 | 0.988 | 0.026 | $2.07 * 10^{-5}$ |
| Student/w.BoH | 0.85 | 0.74 | 0.79 | 0.66 | 0.56 | 0.71 | 0.718 | 0.093 | 0.985 |
| Other Students | 0.69 | 0.62 | 0.68 | 0.47 | 0.25 | 0.60 | 0.552 | 0.153 | 0.135 |
| GPT3.5 | 0.54 | 0.61 | 0.53 | 0.47 | 0.25 | 0.43 | 0.472 | 0.114 | 0.487 |
| GPT3.5/w.FS | 0.69 | 0.55 | 0.53 | 0.41 | 0.50 | 0.60 | 0.547 | 0.086 | 0.987 |
| GPT4/w.FS | 0.69 | 0.64 | 0.60 | 0.77 | 0.63 | 0.66 | 0.665 | 0.054 | 0.518 |
| **Ours** | **0.85** | **0.91** | **0.80** | **0.81** | **0.75** | **0.88** | **0.833** | **0.053** | 0.955 |

The updated dataset table presents a comprehensive statistical analysis of performance scores from various groups, including Human Experts, Students with and without Book of Humanities (BoH), and different versions of GPT models. The Human Experts group exhibits nearly perfect scores with an average of 0.988 and a minimal standard deviation of 0.026, although their scores do not follow a normal distribution. In contrast, the Student groups show more variability, with averages of 0.718 and 0.552 for Students with BoH and Other Students, respectively. The GPT models display a progression in performance from GPT3.5 to our approach with GPT4, where the latter achieves an impressive average of 0.833 with a standard deviation of 0.053, showing a more consistent performance (normality p-value = 0.955).

## F    INTERVIEW WITH HUMAN EXPERTS

We also extensively surveyed human experts about their opinions on our dataset. All of our human experts, who are either university professors of philosophy or PhD students in the humanities, find this dataset both intriguing and valuable. Representing a bridge between traditional academic studies and the latest technological advancements, our application offers a novel method for integrating these two fields. One of our interviewees said, "Given the vast scope of work that no individual could complete in a lifetime, the use of language learning models now makes this formidable task feasible." Another interviewee recognized the philosophical implication of our approach: "Philosophy is a strange field, with a style of inquiry sometimes behaving like mathematics and sometimes like literary studies. The seeming incompatibility between the two sets of assumptions is what keeps me coming back to it, and this investigation clarifies a lot." One professor was intrigued by how our approach gives concrete guidance for practical pedagogical tasks like designing syllabus and creating analytical assignments by showing the interrelations among texts. A PhD student pointed out that the granularity of the information in the dataset is "just right"; the dataset provides crucial clues to interpretation and further learning, without reductive summaries that may discourage students from reading the actual texts.

### F.1    FURTHER IMPLEMENTATION AND ANALYSIS OF OUR DATASETIN PHILOSOPHY

#### F.1.1    SEMANTIC DISTRIBUTION ACROSS REFERENCE TYPES

| Type/Semantic | Nominal | Thematic | Verbal |
|---|---|---|---|
| Negative | 927 | 369 | 134 |
| Neutral | 7923 | 2713 | 1013 |
| Positive | 1376 | 420 | 184 |

Table 6: Distribution of semantic types across reference categories.

#### F.1.2    SEMANTIC DISTRIBUTION IN INTERTEXTUAL FUNCTIONS

| Intertextual Function/Semantic | Negative | Neutral | Positive |
|---|---|---|---|
| Name-dropping | 514 | 6537 | 778 |
| Contextual Explanation | 284 | 2626 | 657 |
| Critical Engagement | 620 | 2361 | 394 |
| Conceptual Application or Expansion | 12 | 119 | 145 |

Table 7: Distribution of semantic types across intertextual functions.

Comparing the network of shared positive references in Fig. 12a with that of the negative ones in Fig. 12b, we find that these philosophers express amicability more overtly and more frequently. They also demonstrate more consensus in their positive acknowledgments of others' work. Our network allows us to compare any of the two philosophers with each other, as shown by Fig. 12c, where we may identify previously unknown relationships. In this circumstance, while Russell and Faguet are rarely discussed together in philosophical discussion, their shared strong sentiment for Homer and against John Stuart Mill cast light on their comparability. It further proposes possible incompatibility



Figure 11: Pie charts showing the distribution of reference types, sentiment types, and intertextual functions.

between Homer and Mill, due to which the commitment to one's stance entails the rejection of the other's. Moreover, by statistically presenting the proportion of each authors' attitudes in Fig. 12d, we identify possible similarities in the tones of their writings. For instance, Jellinek and Oppenheimer may share a more placid style, while Russell's writing tends to be more aggressive.

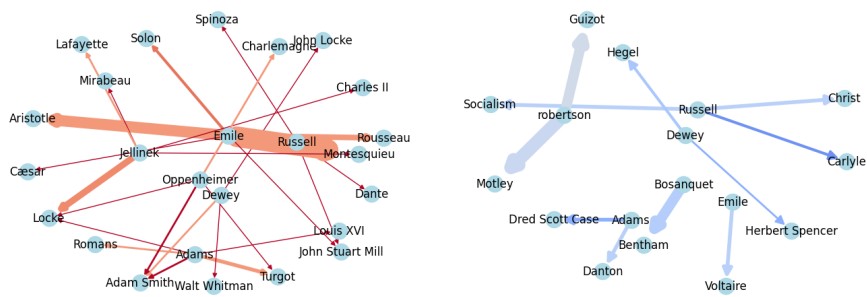

(a) Relationship networks for shared references with positive sentiment score

(b) Relationship networks for shared references with negative sentiment score

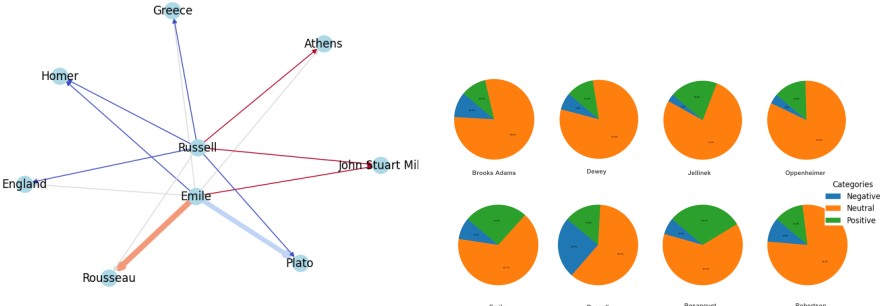

(c) Relationship networks for shared references of Emile Faguet and Russell

(d) Pie chart summary for sentiment

Figure 12: Network and analysis of the authors.

## G    DATA FORMAT FOR FINE-TUNING

To illustrate the utility of the proposed dataset in natural language processing and data science, a sentiment classification dataset containing 2,236 entries has been developed. Each entry includes a sentence from philosophical texts, accompanied by the author's expressed sentiment towards the referenced content within that sentence, as follows:

```
{
    instruction : Please rate the current work sentiment toward 'Montessori system' and characterize the sentiment in terms of
                  negative, neutral, positive
    context : Those educational theorists who have
              had a knowledge of children, such as the inventors of Kindergarten and
              the Montessori system,[14] have not always had enough realization of
              the ultimate goal of education to be able to deal successfully with
              advanced instruction.
    response : Neutral
    category : closed_qa
}
```

Figure 13: Data format for fine-tuning.

## H    TRAINING DETAILS FOR SENTIMENT CLASSIFICATION

The sentiment classification fine-tuning runs based on Transformer package under Python 3.9, where the version of Pytorch is 1.12. All models are downloaded from Huggingface, pre-trained on sentiment or emotion corpus [2].

**Data split:** The dataset is split into training set (70%), validation set (20%), and test set (10%) with the random seed 42 and shuffling. Specially, for BERTweet, the maximal length of each input sample is truncated to 128 due to the fixed model input dimension.

**Hyperparameters:** To reduce the computational cost of LLM fine-tuning, we adopt Low-Rank Adaptation (LoRA) Hu et al. (2021) by Parameter Efficient Fine-Tuning (PEFT) package. For fine-tuning, we adopt Transformer Package. Both hyperparameters of LoRA and fine-tuning keep the same for all experimented models, recorded in Table 8. The hyperparameters corresponding to each model follow the default settings on Huggingface.

The rank $r_{\text{LoRA}}$ is set to 8, determining the rank of the low-rank matrices used by LoRA. It affects the reduction in model parameters and computational efficiency by defining the dimension of the introduced low-rank matrices. The scaling factor $\alpha_{\text{LoRA}}$ is set to 32, controlling the scaling size of the adaptation matrices during training. By adjusting this factor, the magnitude of the adaptation matrices' updates can be balanced to avoid excessively large or small updates. The dropout rate $\delta_{\text{LoRA}}$ is set to 0.1, meaning that 10% of the neurons will be randomly dropped during training, helping prevent overfitting and enhances the generalization capability of the model. Last but not least, the particular modules $\theta_{\text{LoRA}}$ are specified to be fine-tuned. These hyperparameters work together to optimize the application of LoRA in specific models and tasks, balancing computational cost and model performance.

In terms of fine-tuning, the learning rate $r$ is set to 1e-4, determining the magnitude of updates to the model parameters at each step. A smaller learning rate ensures that the model updates its parameters in small, precise steps, contributing to a stable and refined training process, reducing the risk of instability from large parameter changes. The training epoch $E$ is set to 100 to avoid under-fitting but might lead to over-fitting. To help with it, the weight decay rate is set to 0.01 by reducing the size of the model weights at each update. The batch size $B$ is set to 16 due to both the size of our proposed

---

[2]BERT: https://huggingface.co/google-bert/bert-base-uncased;
ALBERT: https://huggingface.co/tals/albert-xlarge-vitaminc-mnli;
BERTweet: https://huggingface.co/cardiffnlp/bertweet-base-sentiment;
RoBERTa: https://huggingface.co/cardiffnlp/twitter-roberta-base-sentiment;
XLNet: https://huggingface.co/TehranNLP/xlnet-base-cased-mnli;
Llama 2: https://huggingface.co/Mikael110/llama-2-7b-guanaco-fp16;
Llama 3: https://huggingface.co/RLHFlow/ArmoRM-Llama3-8B-v0.1;
Mistral: https://huggingface.co/weqweasdas/RM-Mistral-7B;
GPT-2: https://huggingface.co/michelecafagna26/gpt2-medium-finetuned-sst2-sentiment.

sentiment classification dataset and our hardware limitation. Additionally, the optimizer is ADAM, and the load accuracy is 32 bit for all models.

Table 8: Hyperparameters details.

| Module | Parameter | Parameter description | Value |
|---|---|---|---|
| LoRA | $r_{\text{LoRA}}$ | The rank of LoRA matrix | 8 |
| | $\alpha_{\text{LoRA}}$ | Scaling factor of LoRA matrix | 32 |
| | $\delta_{\text{LoRA}}$ | Dropout rate | 0.1 |
| | $\theta_{\text{LoRA}}$ | Modules to be fine-tuned | If XLNet: [layer_1, layer_2] elif Llama or Mistral: [q_proj, k_proj, v_proj, o_proj, gate_proj, up_proj, down_proj] elif GPT-2: [c_attn, c_fc, c_proj] else: [query, key, value, dense] |
| Fine-tuning | $r$ | Learning rate | 1e-4 |
| | $E$ | Training epoch | 100 |
| | $\gamma$ | Weight decay | 0.01 |
| | $B$ | Batch size | 16 |

# I    COMPUTATIONAL RESOURCES

All data collection processes and fine-tuning experiments are conducted on a server with 8 NVIDIA GeForce 3090 GPUs, each of which has 24G memory. The CUDA version is 11.5.

All the resource usage for sentiment classification through fine-tuning is presented in Table 3, including the model parameter count, the proportion of fine-tuned parameters to the total parameter count, and the time required for 100 epochs of fine-tuning. For details on the fine-tuning parameters, please refer to Table 8.

## J  SUPPLEMENTARY ANALYSIS ON SENTIMENT CLASSIFICATION

The confusion matrices of each PLM or LLM is shown in Figure 14. It can be observed that both PLMs and LLMs tend to output a specific class, as seen in the following patterns: Neutral - BERTweet, RoBERTa, XLNet, Llama 2, GPT-4; Positive - BERT, ALBERT, Llama 3, Mistral, GPT-2. Notably, none of the models consistently favors the Negative class, even though Negative samples are the most abundant in the test set. This tendency could be attributed to the differences in the pre-training corpora and methods used for each model. Additionally, LLMs exhibit more moderate biases compared to PLMs, especially in more recent models like Llama 3, which also has the largest number of parameters. This can be attributed to the enhanced language understanding capabilities of LLMs, driven by their larger parameter counts and more extensive training corpora. Nonetheless, this highlights a significant issue: even the most advanced language models suffer from severe mode collapse when directly performing sentiment classification in a philosophical context. Therefore, the most straightforward approach to enhance a language model's understanding of philosophical texts is fine-tuning.

After fine-tuning, it is evident that all models become more inclined to output Negative. To some extent, this suggests that the overall trend brought by fine-tuning is benefiting. However, this trend appears to be extreme, even impairing the models' ability to correctly classify Neutral and Positive instances. This could be due to the imbalance in the training dataset. Similarly, the output bias in LLMs remains less pronounced than in PLMs, which can once again be attributed to the ability of LLMs to better handle imbalanced datasets due to their larger parameter counts.

GPT-4 demonstrates the most stable and balanced performance. Although GPT-4 initially leans towards Neutral, after few-shot learning, it shows improvement in predicting all three classes rather than favoring one. This may indicate that our corpus has greater potential when used for few-shot learning, perhaps even more so than for fine-tuning.

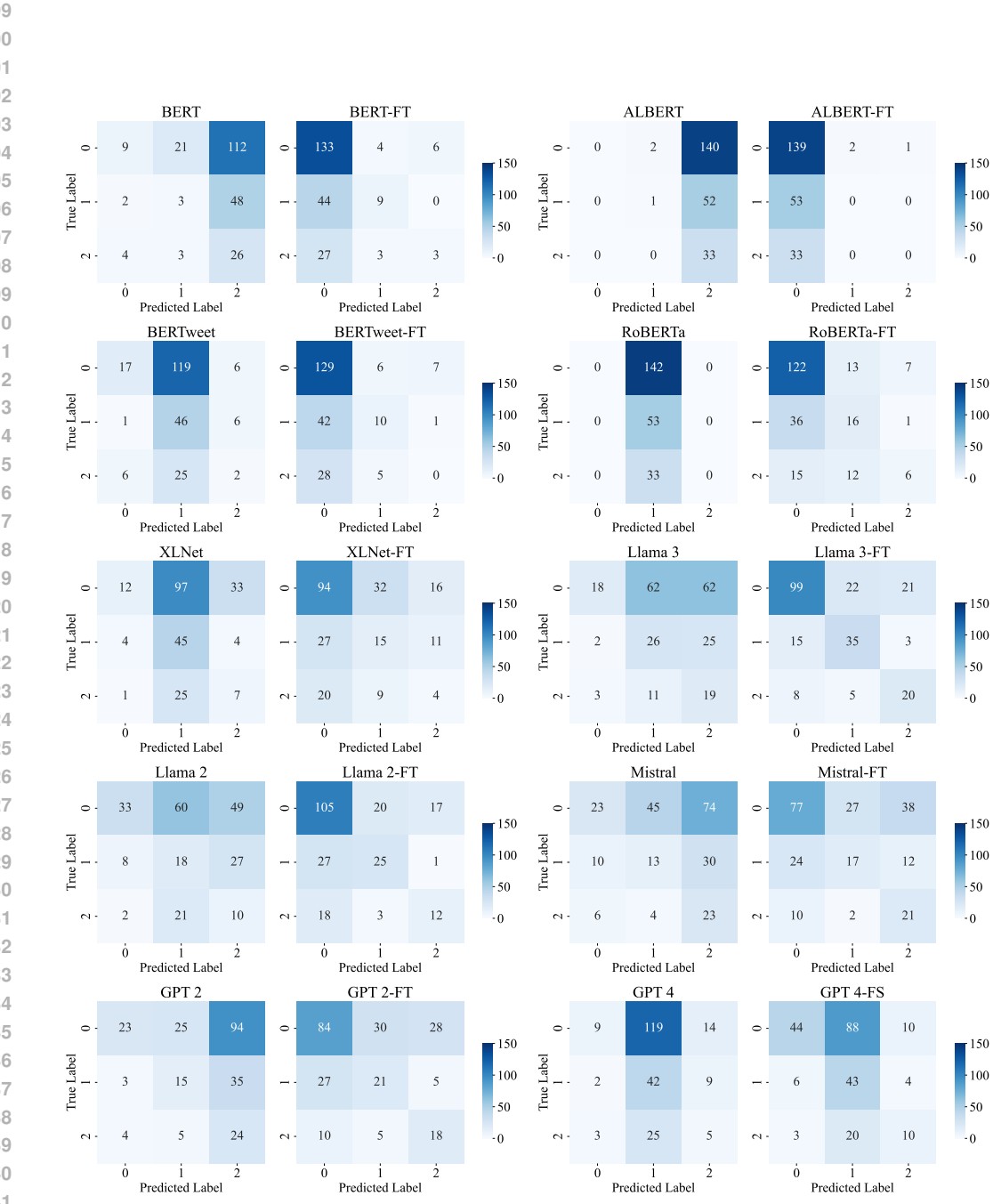

Figure 14: Confusion matrices of each model adopted for sentiment classification before and after fine-tuning or few-shot learning.

