# OpenReview forum: "InterIDEAS: An LLM and Expert-Enhanced Dataset for Philosophical Intertextuality"
_ICLR.cc/2025/Conference — ICLR 2025 Conference Withdrawn Submission_

### Official Review · Reviewer_uDRh · 2024-10-30

**Soundness:** 2
**Presentation:** 2
**Contribution:** 1
**Rating:** 1
**Confidence:** 3

**Summary:**

The paper claims that intertextuality (i.e., recognizing and establihing references to concepts from other texts) specifically in philosophical texts is an interesting application problem for NLP whose mastery translates back into benefits for NL.

The paper constructs a corpus of philosophical texts by integrating manual and LLM-based document selection, and evaluates
the resulting corpus with regard to its quality in two different ways.

**Strengths:**

The paper raises an excellent question -- intertextuality is definitely a largely unsolved problem for language understanding models.
The corpus may also be an interesting contribution to the scientific community in general (unfortunately not for ICLR, see below for details).

**Weaknesses:**

1. This paper is submitted to the wrong conference. To my understanding, ICLR is primarily a machine learning conference looking for technical or at least broadly methodological contributions. The current paper proposes a language understanding *task* and describes a *corpus* but makes no strong original methodological or technological contribution -- the technologies used for the corpus creation are all pretty well understood. I believe that the paper would be a much better fit for a venue in NLP that values corpora as a contribution, such as the *CL conferences. Even there, the short conference paper format may however be problematic in terms of the details needed.

2. The paper overstates the novelty and relevance of intertextuality. Yes, intertextuality is a major problem in philosophy, but this is not as unique as it sounds in the first paragraph: it is also central to understanding in other areas of the humanities, including literary studies (as is acknowledged in the paper, Section 2). Similarly, I am wondering why the semantics of modifiers are surprising to the authors (l. 52/53) -- this has been known in semantics (and philosophy of language) for a long time. Finally, I cannot easily follow the claims in l. 62-68: it is hard to believe for me that LLMs that read philosophical texts (rather than other texts) would quasi-magically improve their reasoning and sentiment analysis capabilities. This cannot be a serious claim - or maybe I misunderstand its scope.

3. The major contribution of the paper, in my eyes, is the corpus construction, but a number of really fundamental questions remain unanswered, maybe due to the limitations of the paper format; but the usefulness of the corpus remains extremely hard to grasp without answers to these questions.
I. What population do the documents with philosophical texts come from? Some kind of corpus? Why is that corpus then by itself not already a good basis for intextuality research? On the other hand, what evidence can you adduce that this population is sufficient for investigations for the research that you envisage? Are key texts guaranteed to be included?  What criteria are even important to evaluate such a corpus for these types of research questions?
II. By including only texts that are already in digitized form, do you not introduce bias towards more accessible / more established works? What about other forms of bias? Or is bias OK in the sense of a 'guided' corpus construction?
III. The same questions, but asked about the InterIDEAS corpus that results from the selection procedure. In addition: what evidence can you adduce that interIDEAS is a *representative* sample of philosophy research in the sense that observations/analyses on InterIDEAS tell you something true about the body of philosophical research at large? Or if this is not your evaluation criterion, what is it then?
IV. What other aspects of quality are relevant when creating a corpus of philosophical writings? How do they compare to "standard" criteria for linguistic or literary corpora?
V. In the evaluation in Tab 1, is is rather hard to understand what exactly the task is: is this P and R relative to a gold standard created by human experts, or some abstract gold standard given by the set of all relevant documents from the population, or something else? More information is needed here. Since this is in essence an Information Retrieval task, maybe adopting best practices from IR might be a good idea here.

4. In the evaluation in Section 5.2 (sentiment analysis) -- if I understand this correctly -- models are fine-tuned on InterIDEAS and then evaluated on "reference-attitude pairs from our dataset" (l. 435/436). In other words, the test data is taken from the data used for fine-tuning. No wonder that the models' performance improves with this fine-tuning step: if my understanding is right, this experiment does not tell us a lot about the quality of the corpus. I would recommend to hold-out a test set not used in fine-tuning (ideally even from a specific topic absent from the fine-tuning dataset) for this type of evaluation.

**Questions:**

See point 3 in Weaknesses above.

---

### Official Review · Reviewer_kYTS · 2024-10-31

**Soundness:** 3
**Presentation:** 3
**Contribution:** 2
**Rating:** 3
**Confidence:** 2

**Summary:**

The paper presents a new large dataset of intertextual references across philosophical texts spanning 200 years (1750-1950) and considering over 3150 writers. The work has been carried out leveraging upon a LLM pipeline and carefully evaluated with a comparison between both experts and non experts on the topic. Finally the authors present two applications of the dataset for philosophical and NLP research.

**Strengths:**

I have found the paper extremely interesting from a dataset construction point of view and it is absolutely an excellent example of fruitful research in between humanities and computer science. Especially the evaluation section, presenting different levels of human expertise, is such an interesting way of highlighting LLMs performance that should be widely adopted for these types of applications.

**Weaknesses:**

The main weakness for the paper is the fact that I believe this work does not fit very well with ICLR as a conference overall. I agree that the focus of the paper is creating a dataset, but I believe the way the ICLR community considers "datasets and benchmark" and how this paper presents it are very different from each other. This piece of work would be an excellent contribution to a Digital Humanities (e.g. Computational Humanities Research), Computational Linguistics (Coling or LREC) or Digital Library (JCDL) conference and would have center stage there. I am afraid that it would instead be a mismatch with ICLR and I believe the authors do not argument this strongly enough why the paper should be part of a deep learning conference (instead of a digital humanities or corpus linguistics venue).

**Questions:**

I would ask the authors to highlight more clearly why they think this paper should be part of ICLR - their final "Application" section focuses generally on NLP applications, but given ICLR is a conference focused on deep learning specifically, what would this paper provide to the deep learning community?

---

### Official Review · Reviewer_Uweh · 2024-11-05

**Soundness:** 1
**Presentation:** 2
**Contribution:** 2
**Rating:** 3
**Confidence:** 4

**Summary:**

This submission examines the use and workflow in construcing an intertextual corpus of philosophical texts from a large, underlying corpus of text.  The authors use a prompted LLM, with a human in the loop (HitL) paradigm to construct the cross-references.  The authors judge the quality of the output against a panel of human annotators, taking humans with advanced humanities degrees as experts.  The results indicate that their workflow improves over baseline LLMs.

**Strengths:**

* Application of LLMs to philosophical texts is an understudied problem.
* Shows a workflow that is applicable to corpus construction for cross referencing.
* Has a side evaluation on the sentiment analysis of the intertextual references.
* Open sources the code and corpus on the anonymous link provided in the frontmatter.

**Weaknesses:**

* More a contribution for application of current LLMs to disciplinary work than a core contribution to learning representations (this conference).  For this reason, I think the submission is a bit outside the scope of this conference.
* Uses standard prompt engineering techniques with few-shot demonstrations, which has already been previously well-demonstrated, so it does not constitute a core contribution.
* Since the workflow uses humans in the loop, the comparison is not particularly fair.  Also, without a clear delineation of the potential novelty to judge the work, it was not clear whether the work wants to be compared with respect to efficacy or not.
* Diagrams have text that is too small to read (e.g., Figures 2 and 3).
* Parts of the evaluation weren't clear.  For example, you evaluate against recall but it was not clear how the authors calculated this.  Was it done in a pooled evaluation style (a la Cranfield style IR evaluations) or some other form of ground truth?
* As applied to Digital Humanities, it would be good to relate the metadata format to other metadata initiatives (e.g., TEI) common to DH.  What do the authoring teams adopt or are influenced by in creating their metadata format and how does it crosswalk to other metadata formats for cross-referencing?
* The sentiment analysis portion of the work did not seem to be very well connected to the motivation of the work.
* The paper wasn't centrally clear whether their workflow or their corpus was the centre of innovation for the paper, leaving it ambiguous and potentially hard to clarify the novelty of their work.

**Questions:**

* It was not clear from Fig 2 whether the prompts are combined into a single prompt or whether the authors expect (most of the) prompt to be reused each time.  Prompt gisting might be a useful technique to reduce the inference cost (although that may be automatically done by the underlying backbone LLM system, especially if a commercial system is used).

- (More of a comment) I really want to accept this paper, as I think it does add value so that other users can share good use cases and validation methods, but I feel that this is the wrong venue for such work.  Other DH conferences (such as JCDL or DH) seem more appropriate for this kind of work.  ICLR is more for practitioners working in the core area of AI and technological innnovation.
- (Also a comment) Since part of the motivation of the work is the variety of intertextual references, it would be good to see more of a dissection of the error types and efficacies of the LLM on these forms of implicit references.

- 9, 434: the authors created a corpus of reference--attitudes, but there is no details on its construction, including the interannotator agreement, and whether an AI system was used to create the reference data, which might indicate a source of bias.
- 10, 491: I disagree that the reason for the performance shortfall is due to mode collapse.

**Details Of Ethics Concerns:**

Bias coming the inclusion / exclusion filters and how well the LLMs are able to find the intertextual references and sentiments should be further expounded on.  To be clear, the authors have already done this to a fair extent, so I don't think ethics review is necessary but algorithmic and automation biases should be highlighted.

---

### Official Review · Reviewer_zHEp · 2024-11-06

**Soundness:** 2
**Presentation:** 2
**Contribution:** 2
**Rating:** 3
**Confidence:** 4

**Summary:**

This paper introduces a framework using Large Language Models (LLMs) to determine citation contexts and intents in philosophical texts. This framework facilitates philosophical analyses, such as examining the interconnections between different philosophical schools over time. Leveraging this framework, the authors have extracted over 15,000 citation relationships from a corpus of more than 45,000 pages of modern philosophy texts in English.

**Strengths:**

Below, I outline what I consider to be the primary strengths of the paper.

1. This paper appears to be one of the first studies (to the best of my knowledge) focusing on the analysis of citation contexts and intents within philosophical documents.
2. The dataset curated and utilized in this study offers a resource for philosophers, enabling them to comprehend the interconnections among a large volume of documents more efficiently and effectively.

**Weaknesses:**

Below, I outline the main weaknesses of the paper.

The paper addresses the identification of citation contexts and the classification of citation intents, topics that are already well-established within the field of Natural Language Processing (NLP) [1]. Unfortunately, the paper does not contribute new insights, as existing NLP frameworks are sufficiently robust to be applied directly to philosophical texts without significant modification.

[1] Measuring the Evolution of a Scientific Field through Citation Frames (Jurgens et al., TACL 2018)

**Questions:**

I would like to pose the following questions to the authors:

1. Could you explain the choice of using accuracy and recall to evaluate annotator agreement rather than employing more conventional measures like Fleiss’ kappa, which are typically used for assessing inter-annotator agreement?
2. Recognizing that new philosophical ideas often build upon older ones—a common phenomenon in scholarly work—could you highlight the principal discoveries presented in section 5, aside from this?

---

### Note · Authors · 2024-11-15

**Comment:**

Dear Reviewers:
We wish to extend our heartfelt thanks for the insightful comments and valuable suggestions we received during the review process. Your feedback has greatly contributed to our ongoing research efforts.

After thoughtful consideration and taking into account your suggestions, we have decided to withdraw our paper from consideration at ICLR. We agree with your assessment and believe that the focus and contributions of our work may be more appropriately suited to a different venue that aligns more closely with the specific topics of our research.

Thank you once again for your time and the constructive critique that has helped steer our research in a positive direction.

Best,
All Authors

**Withdrawal Confirmation:**

I have read and agree with the venue's withdrawal policy on behalf of myself and my co-authors.